# Promoting Women’s Mental Health and Resilience in Times of Health Crisis and Adversity via Personal Development Groups

**DOI:** 10.3390/healthcare13091035

**Published:** 2025-04-30

**Authors:** Maria Moudatsou, Areti Stavropoulou, Michael Rovithis, Dimitrios Mimarakis, Sofia Koukouli

**Affiliations:** 1Department of Social Work, Faculty of Health Sciences, Hellenic Mediterranean University, 71410 Heraklion, Greece; mimarakisdimitrios@yahoo.gr (D.M.); koukouli@hmu.gr (S.K.); 2Laboratory of Interdisciplinary Approaches for the Enhancement of Quality of Life (QoLab), 71410 Heraklion, Greece; astavropoulou@uniwa.gr (A.S.); rovithis@hmu.gr (M.R.); 3Department of Nursing, Faculty of Health and Care Sciences, University of West Attica, 12243 Athens, Greece; 4Faculty of Health Science, Social Care and Education, Kingston University, London KT2 7LB, UK; 5Department of Business Administration and Tourism, School of Management and Economics Sciences, Hellenic Mediterranean University, 71410 Heraklion, Greece; 6Social Service, Pammakaristos Hospital of Divine Providence, 11144 Athens, Greece

**Keywords:** personal development group, group therapy, women’s empowerment, personal growth, women’s resilience

## Abstract

**Background:** Women carry out an array of demanding tasks due to their multiple roles as mothers, workers, spouses, and caregivers. Their responsibilities to the family and society are essential throughout life, but they become even more important at times of crisis and unanticipated events. All these obligations may have a detrimental effect on their mental health and general well-being. According to the social model of health, through personal development groups, women improve their health because they can transform their personalities, enhance their social and personal abilities, and strengthen their resilience to unforeseen occurrences, health crises, and adversity. **Aim of the study:** This follow-up study examined the viewpoints of women regarding the durability and efficacy of group therapy’s positive impact on their resilience and mental health, especially through unforeseen circumstances and crises. This group of women had taken part in group therapy six years ago to strengthen their resilience and improve their mental health. **Methods:** Data collection was implemented through semi-structured in-person interviews conducted from December 2020 to March 2021. Five topics emerged from the framework analysis: (a) personal development and mental health; (b) reinforcement of their resilience; (c) group therapy and women’s health during the COVID-19 pandemic; (d) an assessment of group therapy on women’s health through imagery; (e) future recommendations. **Results:** Most of the women stated that group treatment had a beneficial impact in terms of improving their individual abilities. The group experience was described as a bridge that allowed them to recognize and accept their emotions. Since COVID-19, women have developed helpful coping strategies to deal with daily tension, loneliness, and work-related stress. **Conclusions:** Our research results indicate that group therapy is a useful tool for women’s empowerment and mental health in times of crises and adversity. Social policy should take it into account in order to meet women’s highly demanding roles and personal needs.

## 1. Introduction

Due to their multiple tasks as caregivers, mothers, wives, and employees, women have an important role to play in the family and society throughout their lifespans and during unexpected events. Worldwide, women’s autonomy continues to be severely restricted in a variety of circumstances. Compared to males in the same nations and women in wealthier countries, women in low- and middle-income countries (LMICs) often report lower levels of life satisfaction, control over their life, freedom of choice, and having a voice in household decision-making [1].

Socio-cultural factors that impact women’s lives and health relate to unequal power dynamics between males and females, social norms linked to lower levels of education, and paid employment opportunities. Furthermore, concentrating solely on women’s reproductive roles and their real or future experiences of sexual, emotional, and physical abuse may seriously impair their physical and mental well-being [2]. Women’s lives are related to gender-based perspectives and should be explored under these social, cultural, psychological, and gender norms [3].

Socially constructed limitations, which have been a cause and an effect of women’s inability to advance and accomplish their goals, have worsened their mental health. As a result, they experience to a greater degree negative emotional influences, have a lower perceived level of self-confidence, and have lower aspirations for progress [4].

Personal growth initiative (PGI) is a fundamental component of mental health and self-development and a sign of one’s desire for autonomy. It includes four dimensions: preparation for change, planning, use of resources, and conscious actions [5]. PGI indicates an individual’s purposeful and active involvement in their own improvement. It initially emerged as a potentially helpful therapeutic concept. Personal growth happens when individuals are involved in the therapeutic process and have an active role in their personal improvement [6]. It consists of both cognitive (readiness for change) and behavioural (an individual’s usage of external resources and intentional behaviour) elements [6].

Personal growth is positively related to emotional and mental health and well-being [7,8,9], while it is negatively associated with distress and depression [7]. It may happen in various settings, but therapeutic procedure has a great influence on it [10]. Throughout the therapeutic process, clients are enabled to be involved in their personal development [6]. Personal growth is making the most of one’s distinctive strengths and abilities in order to accomplish objectives [11]. Those who have a grown personality take responsibility for their life. They demonstrate appropriate behavior and restraint by acknowledging the unfavorable aspects of their existence [12].

Personal development groups (PDGs) are widely used in psychology, psychotherapy, social work, and counselling, since it is believed that they promote both professional and individual growth [13]. Personality development constitutes a personal and collective life experience that affects people’s social roles and environments [14]. In PDGs, the concept of individual empowerment is central. It is related to increasing one’s personal power, self-esteem, optimism, and hope.

Personal development groups have been described in the literature with a variety of terms, such as growth groups, reflection groups, and personal development/growth groups [13]. They focus on self-discovery, interpersonal interactions, and individual development [13]. They are frequently utilized to increase the self-awareness and personal growth of their members [15]. Challenge, feedback giving and receiving, member disclosure of oneself, and investigation are all elements of these groups [13].

PDGs provide their participants with improved leadership, listening skills, awareness of other cultures, deeper self-awareness, experiential learning about vulnerability, self-disclosure, and acceptance [16]. They increase their positive influence on their members when they have mediators with the appropriate abilities. These mediators give accurate instructions about the goals, potential risks, and benefits of the group; assist group members in setting personal goals; provide education regarding self-disclosure; avoid peer pressure in the group; and schedule follow-up individual sessions with group members [17,18].

Women who participate in a personal development group for their individual and social empowerment can transform their personalities and become more resilient to any type of crisis, disaster, or unexpected event [15,19]. Throughout the COVID-19 health crisis, women reported more detrimental effects on their mental health than men. They experienced high levels of stress, loneliness, sleep disturbances, depression, symptoms of post-traumatic stress disorder, and sadness [20]. Group or personal therapy might reduce their isolation and enhance a sense of mutual support. They both increase positive feelings linked to optimistic individuals who perceive hurdles to achievement as challenges to overcome rather than as causes of stress [21]. Because of their therapy, women managed to cope with their unpleasant experiences, trauma, and disastrous events. They started to reconstruct their own growth via the cognitive and emotional struggles that followed the trauma, which might result in adjustment and well-being [22]. The social model defines health as individuals’ total physical, mental, and social well-being, and not only the absence of disease [22]. According to parameters of internal social determinants of health, women’s psychological and mental health is improved by self-development, personal empowerment, self-confidence, emotion handling, coping strategies against crisis, and the reinforcement of resilience [22].

People do not live in isolation. They interact with their social and environmental contexts. Resilience is a protective factor against stress and anxiety created by negative life circumstances. It is defined as a dynamic process wherein individuals display positive adaptation despite experiences of significant adversity or trauma. This term does not represent a personality trait or an attribute of the individual. It rather stems from the interplay of internal and external resources. It has individual, social, and cultural perspectives [23,24]. The ecological approach defines resilience as a dynamic interplay between personal (active coping, reflective functioning, meaning-making, optimism, flexibility, and hopefulness), relational (social support, partner relationships, and family relationships), and contextual (health and social networking, cultural factors, and social policies) factors [23,24]. Therefore, when exposed to adversity, people will draw on individual resources but also interact with their environment by drawing on community and societal resources which may facilitate (or hinder) resilience. The precise role that these factors play in fostering resilience is, in fact, unknown.

## 2. Methods

### 2.1. Aim

The objective of the current follow-up study was to evaluate the perspectives of women who had undergone group therapy from 2012 to 2015 regarding the efficacy of the group in improving their mental health and enhancing their resilience, especially during times of unforeseen circumstances and crises. In our study, women’s resilience and mental health was evaluated six years after the completion of group therapy. Therefore, other factors, such as the women’s social and contextual environment, might have had an influence on these positive outcomes. To address the knowledge gap regarding the long-term impact of personal development groups, the same interview guide was applied as the one used during the initial assessment of the group influence. Additionally, a brief reminder of the thematic units covered in the group was provided.

### 2.2. Description of the Group Therapy

The goal of the group therapy was to help women develop their social and personal skills and meet their psychological and social needs, stemming from the multiple family roles and the newly emerging social conditions. The sample’s participants at the time were chosen using a deliberate sampling procedure. A purposeful sampling strategy involves choosing research subjects who are knowledgeable about and experienced with the phenomena being studied, as well as those who can respond to the study’s objectives. Because the researcher uses their own judgment to choose the better-qualified study participants, this non-random sampling method is also known as judgmental sampling. Using this method, the researcher chooses out of the population under study those individuals who fit the study’s parameters and are able to respond to the research questions sufficiently and in depth. This indicates that the selected participants are highly informed on the subject of the study and are able to provide a wealth of information about it [25].

This group therapy took place in a municipality in the Greek island of Crete. Access to the study population was gained through the help of the heads of the civil service and the social services.

The group therapy took place in three semesters between 2012 and 2015 and consisted of twenty meetings throughout each semester. Each meeting lasted two hours. Twelve women aged between 35 and 45 years were involved in the group, and the therapy methods used were relaxation techniques and art therapy methods, such as guided visualization and guided imagination. These methods enable people to come in touch with the deeper, less known and processed depths of their inner self. This way they have more possibilities to gain self-awareness [13].

Indicative topics for discussion were collaboration, communication, self-esteem, emotional expression, conflict management, the expression of desires, femininity in the participants’ lives, their relationships with their mothers, their priorities, responsibility in life, the management of inactivated abilities, the failures in their lives, and coping with their past experiences.

The interview schedule focused on psychiatric illness concepts derived from the DSM IV [26]. The purpose was to exclude unhealthy habits that need to be closely monitored by a psychiatrist and to reassure them that all of them were eligible to follow the demanding group therapy techniques and rules.

The participants in the group therapy were purposely selected from those who responded to the announcement in the local newspaper. The first author and a family therapist candidate selected them through personal interviews.

To acquire the participants’ consent, detailed information on the study’s scope, as well as its inclusion criteria, were initially provided over the phone and subsequently via email. Both the voluntary nature of the survey and the anonymity of the respondents’ answers were guaranteed. The period of the team’s preparation lasted approximately two months.

### 2.3. The Design of the Present Study

The present qualitative descriptive study lasted from December 2020 until March 2021. Through the use of qualitative research methods, people’s opinions, experiences, perceptions, and emotions may be thoroughly examined [26]. Qualitative descriptive research does not attempt to explain things; rather, it describes them. It strives to offer a thorough account of what happened. Qualitative descriptive research is an appropriate method to acknowledge the subjectivity of the problem and the diverse perspectives of the participants and convey the results using language that either closely matches or directly represents the original research topic [27].

The goal of qualitative studies is to provide answers to who, what, where, and how questions by using a journalistic method [28]. Descriptive qualitative studies are suitable in evaluations and assessments of programmes and interventions because the information gained may be used to create and improve programs or surveys [29].

### 2.4. Participants

The sample of the present study (12 participants) consisted of all the women who had attended the group therapy conducted between 2012 and 2015. Women who expressed a desire to enhance their resilience and meet their mental needs attended the personal development group. It was part of a larger project aiming at the evaluation of women’s health needs at the local community level. Therefore, the inclusion criteria of this study were “having previously attended the group therapy”. Twelve (12) women, who participated in group therapy between 2012 and 2015, constituted the final sample of the present study. They were all married except for one who was divorced. Ten of them had a university degree, and two had completed secondary education. All of them were between 35 and 48 years old. Two participants worked in the public sector, nine worked in the private sector, and one was unemployed (Table 1).

### 2.5. Data Collection

Semi-structured, face-to-face interviews were used for data collection. This method of interviewing is largely used in qualitative research, allowing the researcher to thoroughly explore the participants’ thoughts and beliefs [30].

### 2.6. Measures

The interview guide focused on two main axes: the subjective experience of group therapy for personal development and the future recommendations for group therapy’s improvement. The authoring team developed a semi-structured interview guide, based on the latest literature on group therapy and personal growth. The interview scheme included one introductory open-ended question, which was used at the beginning of the interview to motivate the participants to share their experiences with group therapy. Open-ended questions on the advantages and disadvantages of group therapy and suggestions for future implementation were also included. The interview guide consisted of the following thematic topics: the personal development group and its impact on mental health, the personal development group and resilience, and visual metaphors representing the group’s overall contribution to women’s mental health and resilience. The following are examples of research questions: What influence did the personal development group have on your self-esteem? Are there skills you have gained from your participation in this personal development group? Would you like to talk about them? How do you think that this personal development group helped you in identifying and controlling your emotions? Would you like to give us some examples?

Within the specified timeframe (May–June 2022), all respondents participated in online interviews implemented methodically by the first author (M.M.), who possesses competence in qualitative research methodology. None of the participants withdrew from the study. An interview typically lasted between thirty and forty minutes.

### 2.7. Data Analysis

The data were analyzed using framework analysis. This is a typical method to manage and analyze vast amounts of data from semi-structured interviews. Multidisciplinary research teams utilize it to create a clear and comprehensive description of research data. It is an appropriate data analysis method because it ensures accurate findings by adopting a systematic approach to data management and analysis, creating a clear audit trail throughout the study process. We chose framework analysis for our study due to our diverse team of academics and the vast amount of data deriving from the semi-structured interviews [31]. This approach is composed of five separate but related steps: (a) making the data easier for the researcher to understand and interpret; (b) identifying codes that characterize the information in the data; (c) identifying more general categories and themes; (d) organizing the data according to the concerns raised; (e) writing the results in a way that makes sense and provides context for the data [32]. Ultimately, the study’s trustworthiness was established by the implementation of certain strategies, including reflexivity, peer review, and analyst triangulation. Two members of the study team separately examined the data. Interpretations were contrasted and argued until agreement was reached on the most representative ones. Furthermore, peer investigators—academics with expertise in qualitative research—evaluated the study procedures, and the researchers kept reflective notes about the interview process and the study settings [33,34,35]. Themes were coded and indexed by identifying the key issues and the variety and frequency of experience for each one, after searching for the existence of typologies and taking into consideration all the possible or negative attitudes.

### 2.8. Ethical Issues

Prior to the commencement of the study, ethical approval was gained by the involved organizations (head of the municipality in the island of Crete: ref. no. 790:30-09-2020). Throughout the investigation, proper ethical guidelines were followed, and the human subjects were protected [36]. The participants were thoroughly informed about the purpose and the design of the study prior to the interview commencement. Permission for using a tape recorder was also granted by the interviewees. Participants signed the informed consent forms, and the researchers verified that participation was voluntary, anonymous, and confidential. The women were also assured that the information collected would only be utilized for scientific purposes and that the results would be disseminated without revealing any personally identifiable information. All twelve members of the group therapy agreed to participate in this study as well.

## 3. Findings

Data analysis revealed the following four themes: (a) personal development and mental health, (b) reinforcement of resilience, (c) coping with health crises; (d) assessing group therapy’s impact through an image. The main themes and sub-themes, as well as the key indicators that guided these themes and sub-themes, are presented in Table 2.

### 3.1. Personal Development and Mental Health

#### 3.1.1. Developing Self-Confidence

The vast majority of the women reported the positive influence of the group therapy on enhancing their self-esteem.


*«These days, self-esteem is crucial to my existence. I reaffirmed it to our group. I resisted it. I asserted it. It was difficult to grasp at first. I just learned now. You do nothing at all if you lack self-worth. Everything we do is based on our positive perception of ourselves. I appreciate the group for this»*
(R7)

Additionally, the women illustrated that the group was a source of security and protection that made them trust not only themselves but other people.


*«I learned how to trust myself and others initially. To avoid seeing adversaries wherever»*
(R8)


*«I was able to interact with others because I felt secure in the group. To approach them and not be scared of the intimate personal connection»*
(R5)

#### 3.1.2. Handling Emotions

The group experience was a bridge that gave the women the opportunity to come in touch with their feelings. Most of them were reluctant to uncover their feelings. They were ashamed of expressing themselves because they thought that an expressed emotion was a reason for accusing them of being unable to fulfil their multiple roles.


*«The group was very helpful. I became skilled at quickly and readily recognizing my emotions. I acquired the ability to move through each phase. In particular, it was quite helpful for me in finding ways to regulate my feelings of pain»*
(R1)

Another participant indicated the following:


*«I experienced all of the emotions to the maximum before the team. I was going too far. I put a measure now. I found the rules. I learnt to keeping in touch with my feelings. My being is free of any indication of embellishment»*
(R4)

Another addressed the feelings dimension:


*«Before getting on the team, I always battled with my emotions. I was not sure how to act out and communicate them. I strive to control and communicate them now that I am aware of them. I was a rather secretive person. I now express my emotions outwards»*
(R6)


*«Now permit me to take my bravos. My kids hold me in high regard. They see my value because I’ve come to understand it myself. Both I and others take care of ourselves. I also continue to do the self-esteem strengthening exercises I learned in the group»*
(R7)

### 3.2. Reinforcement of Resilience

#### 3.2.1. Overcome Difficulties

Through group therapy, the women learned to overcome difficulties. They had learnt coping strategies to work and to live under pressure. Some participants in the group felt that the personal development group assisted them in stress management.


*«I developed stress management skills. Not to share with my family and coworkers. But to be more composed. To be free to enjoy without having to worry about anything all the time. To survive. I’m not sure what was more helpful to me. Everything, I believe. I talked, felt trusted, and received assistance»*
(R8)

Another one added the following:


*«My anxiousness was a major concern for me. I was living it. Years at this point. I was having severe gastrointestinal issues. I’ve improved since then. I discovered how to de-stress. To organize things. To establish priorities. Not to accept accountability for anything»*
(R9)

However, there were also views from participants who believed that they had not had enough assistance in stress management.


*«To be very honest, I don’t think I received as much assistance as I would have liked in managing my anxiety. I’ve made some progress, but not as much as I would want. I continue to be quite resistant. My life is still pending this. However, I have trouble. Try it. I’m unable to figure out why I wasn’t assisted now. Perhaps my anxiousness was too great, and maybe I should have allowed it to go longer. I could have spoken up more. I didn’t want to impede the others’ progress with my own worry because I could see they were moving forward with theirs. Now, if I were to rejoin the team, I would request to discuss»*
(R11)


*«I’m still having trouble. I developed a rather poor ability to manage stress. Before, I was more prone to panic. I’m doing better now, too, but not as well as I’d like to. It’s still something I can’t seem to ignore. I would want additional exercises if I were back on the team. I would like to receive more help»*
(R12)

#### 3.2.2. Improvement of the Relationship with Family

All the women supported that the group therapy helped them to assess the financial crisis under a new framework. According to the technique of reframing, the women acquired a systemic response to the crisis. They understood that the financial crisis is a part of the social and economic system’s crisis. The concept of crisis might be considered as a means of setting new priorities in their family life. The financial crisis resulted in deeper relationships and new ways of communicating.


*«The group therapy had a positive association with our family life… last year we were about to collapse due to the financial crisis and the role of mass media… but we manage to overcome our sadness… our family shapes new attitudes, such as finding better ways of coming in touch between each other, rather than putting priorities on expensive clothes and materials… I think that I helped them because I had the group and through our daily interaction and communicating, we explored all the possible parameters of improving our life…»*
(R5)

Group therapy enabled women with new mechanism of coping with the stress and anxiety that derived from the financial crisis.


*«My group experience was very helpful for me and my family… we were taught to take care ourselves and to reduce our stress… we realized that money is not the most important think in the world… we change many of our expensive habits with new ones,… more cheaper… we discover alternative ways of laughing and making fun…… we have not to spend money in everything in our life…»*
(R7)

The great majority of them admitted that group therapy was a means of making improvement in relationships with their families and especially with their mothers. The group acted as a bridge that linked the women of different generations.


*«The group was a great chance for me to realize my mistakes in the relationships with my mother… to find alternatives ways to cope with her… but on the other hand, taught me to improve my relationship with my little daughter»*
(R12)

#### 3.2.3. News Ways of Communicating

The participants found new ways of communicating and coping with interpersonal conflicts.


*«I realized the importance of respecting other people… and… I was taught to hear other people… to hear my husband and my children… I am not suspicious any more…»*
(R11)


*«It is easier for me now to express myself to my family… to admit my mistakes without hesitation… to cope with quarrels… to overcome unimportant issues… to assert my wishes»*
(R9)

### 3.3. Coping with Health Crisis

#### 3.3.1. Self-Efficacy

Participants had the self-efficacy to cope with health crises such as COVID-19. They managed to cope with anxiety and the stress created by the pandemic.


*«Above everything, I remained composed. Although I was afraid, I was able to control it now that I had a team. The processes were familiar to me. I was aware of how»*
(R7)


*«COVID-19 caused me a great deal of stress and anxiety. Without the group, I’m not sure what I would do»*
(R5)


*«Using the coping mechanisms we discussed in the group, I was able to manage my extreme stress»*
(R6)

#### 3.3.2. Strategies to Overcome Daily Difficulties

The participants were calm and found strategies to overcome the daily routines, the loneliness, and the difficulties created by COVID-19.


*«I learned how to make daily objectives from the group. To follow a timetable. Stay engaged and avoid boredom»*
(R8)


*«I managed to stay in touch with my family in spite of the challenges posed by COVID-19. My relationship with my spouse and kids grew closer. Together, we completed tasks and overcame isolation and loneliness»*
(R10)


*«We’ve benefited greatly from technology. We had an instant connection. with acquaintances, coworkers, friends, and family. We were having coffee together over Viber with my pals. Even birthdays were commemorated on Skype»*
(R9)

They found creative ways to improve their knowledge in their work.


*«It was a chance for me to advance my job. To work on some recommendations that I’ve had for a long time. I had the power to transform feelings of loneliness and isolation into action and creation»*
(R12)


*«I was able to succeed and observed the advantages of remote work. I believed it would be catastrophic to skip work. Nonetheless, I succeeded. I recognized the advantages. I used to have more time to get ready. I kept the commute to myself. I concentrated on achieving my objective»*
(R11)

#### 3.3.3. COVID-19 as a Challenge

The participants viewed COVID-19 as a challenge rather than a catastrophe in their lives.


*«It was a chance for me to reflect on my goals and desires»*
(R5)

Additionally, another one suggested the following:


*«I did not consider COVID-19 to be a catastrophe. My perspective was more positive. It’s a crisis that presents the chance to reflect and begin over»*
(R3)

#### 3.3.4. New Ways of Behaving

The participants found new ways of behaving in personal and family life.


*«First, I worried that spending so much time with the family would lead to conflict and friction. However, everyone has personal boundaries. We moved forward because our collective hopes were granted. None of us felt under any kind of pressure, and we had a terrific time»*
(R1)

### 3.4. Describing Group Therapy’s Impact Through an Image

#### 3.4.1. Images: Sense of Improvement and Continuous Development

Most of the women translated into images the positive effect of group therapy on their mental health. Under the concept of guided imagery, they relaxed and talked about the group’s effectiveness. Their images flourished with the sense of improvement and continuous development.


*«The role of the group in my mental health, looks like with a difficult road but it is filled with very beautiful roses… their smelling enables me to keep walking… My group experience is as a pink rose with dewdrops»*
(R4)


*«Group acts as an opened pomegranate that is crowded by its red hoods…»*
(R1)

#### 3.4.2. Images: Source of Power, Support, and Caregiving

All the women accepted the active role of the group in their mental health as a source of power, support, and care giving.


*«It is like a nest with newborn chicks… that provides all of us, with love, care, support… it looks like as a second family for me… that enables me to take care my real family…»*
(R3)

Each member of the group suggested that the group be revived and that participants continue to take part in it.


*«How I wish we could start over as a team. It’s what makes me feel good about it»*
(R2)


*«I want my life among the group back. For me, it was and continues to be a source of support and strength»*
(R4)


*«The group was a great support to me. I’d like it to keep going forever»*
(R6)


*«I occasionally dream that this group is endless. I would adore it. Its support has been crucial for me»*
(R8)

## 4. Discussion

The study sample reported that the personal development group had a positive influence οn their life because of many beneficial enhancements and changes in all parts of their personality. Personal growth is the concept of making the most of one’s unique assets and abilities in order to accomplish objectives [11]. People start to act and behave differently than before as part of their own growth. As a component of personal development, people’s cognitive ability is impacted by their increased capacity for thought and reasoning. Hence, they can adjust their behavioral priorities and different reactions to a situation or relationship [37]. Studies have additionally revealed outcomes that are analogous to the beneficial impact of personal development group therapy [13,37,38,39].

The women of our sample enhanced their mental health through this process. Mental health emphasizes the significance of people’s inner resources in personality development and their behavioral self-regulation [40]. Τhe participants referred to the fact that they were in touch with their needs and that they had used coping strategies in managing their difficulties and their problems. They developed new skills in overcoming their difficulties and reaching their goals. In line with our results, women in Australia were empowered through their participation in group programmes. They improved their mental health via setting objectives, solving problems, and decision-making. They also indicated that they had an active and involved life [41].

During personal and/or social and financial crises, the women of our sample had the self-efficacy to be involved in new tasks, to take responsibilities, and to be familiar with the new challenges in their personal, professional, and social lives. Self-efficacy raises peoples’ persistence in the midst of the anxieties, obstacles, and stress that they experience when coping with issues related to contextual difficulties [42]. Moreover, it gives a person optimism for handling his/her social and personal objectives [43]. Our results are connected with studies suggesting that personal development groups increased personal self-efficacy in psychotherapy students who participated in group therapy [44,45,46].

The women in our study managed their self-development and personal growth. They were taught to have self-awareness and self-improvement. They became aware of their emotional limitations and deficits as a result of this information and experienced an increase in motivation and self-control after learning problem- and emotion-oriented dreaming techniques. People who participate in the growth process are more eligible to raise their skill set, including motivation, behavior, attitude, and cognitive processes [47]. This happens because personal growth initiatives acquired during group therapy enable them to be eligible for self-improvement and self-change [40]. There are studies with similar positive results that indicate the importance of group therapy in women’s emotional intelligence and self-control [13,40,41,48]. Self-development puts a high priority on recognizing feelings and emotions, ensuring the most effective use of methods for problem-solving, raising awareness of everyone’s distinct skills and effectiveness and working to develop them, and providing accurate training [40]. People who participate in the growth process are more eligible to raise their skill set, including motivation, behavior, attitude, and cognitive processes [46].

Our results suggested that the women managed to have a sense of empowerment and mental health throughout the follow-up, six years later, after the COVID-19 pandemic. This is a sign of the stability of their personality development after taking into consideration the fact that the women managed to keep their self-efficacy and mental health improvement. We experienced the same positive results in a relevant study conducted three years prior on the same sample of women [49,50]. Positive follow-up findings demonstrated the wide durability of improvements achieved once psychotherapy was implemented [51].

Additionally, the women of our sample indicated that they managed to be resilient and to have hope during COVID-19. They appraised that group therapy enabled them to overcome the negative psychological impact of the health pandemic. COVID-19 has been referred to as a severe life trauma that has had detrimental effects on people, including symptoms of post-traumatic stress disorder, anxiety, sadness, and terror [21]. Women have been resilient and mentally healthy because they have had the skills and the resources to rebuild their personality and to keep calm despite the stressful life events [52]. Certainly, in our study, there was a gap in the way resilience was built, because it was defined as the capacity for managing the availability and utilization of psychological, social, and physical resources [23]. According to our findings, group therapy only had an impact on psychological or personal aspects. We did not have information about the women’s social and contextual interactions and their role in fostering resilience and promoting general well-being.

Our results indicated that the women members of the personal development group were in a lifelong process of mental health and personal development. They had the inspirations and motivations to develop their personal capabilities and improve their health according to the internal social determinants of health, such as self-development, personal empowerment, developing self-confidence, emotion handling, coping strategies against crisis, and reinforcement of resilience [22]. An explanation might be the therapists’ techniques [19,53]. The therapists made use of experiential techniques and techniques that relied on art forms such as guided visualization, stories, and techniques of relaxation. The therapist especially guided imagery by making use of stories with metaphors and similes in order to give instructions throughout the sessions. Using art, one can bridge both the conscious and unconscious minds and reach higher levels of self-awareness and creative consciousness. Therapists can thus provoke a healing and corrective intervention on the wounded imagination with the appropriate therapeutic interventions [50,54,55,56]. Most of the women translated into images the positive effect of group therapy on their mental health.

## 5. Limitations

This study was subject to some limitations. It was a self-reported assessment and may not have described the actual situation of these women. A recall bias is also possible given the six-year interval between the initial intervention and the follow-up study. Also, the data were susceptible to optimistic bias because they were based on self-reports and the women might have exaggerated the positive influence of the group therapy on their mental health. Moreover, because the researcher and the therapist of the group were the same person, there might have been a social desirability bias. Possibly, this fact prevented the women from giving truthful answers. Undoubtedly, the six-year gap between the last intervention and the follow-up study demonstrated the consistency of the beneficial impact of the group therapy, and that is one of the strengths of the study. Nevertheless, the extent to which the personal development group affected the women’s resilience remains unclear. Resilience is a combination of personal, social, and contextual factors. Although we thoroughly examined every aspect of resilience during each intervention to identify which components of the women’s resilience were the result of the group intervention, there are still issues. The fact that the women’s self-reports were the primary means of cross-checking the accuracy of our findings raises questions about the group’s impact on women’s resilience.

## 6. Conclusions

This study underscores the importance of group therapy for women’s life. Our results highlight key factors about the role of group therapy in fostering women’s’ mental health and resilience. A possible explanation is that group therapy activates mechanisms and processes that enhance an individual’s self-esteem and self-confidence. Furthermore, it contributes to the development of new skills in communicating, handling emotions, solving personal problems, and coping with stress. It enhances resilience through active coping, optimism, and hopefulness. In addition, our findings add to a growing body of literature related to the contribution of group therapy in coping with the COVID-19 health crisis.

Group therapy is an effective therapeutic approach that social workers, psychologists, psychotherapists, and other practitioners can utilize in combination with individual or family consulting. At both the local and national levels, social policy plays a vital role. It can cover training needs and provide funding to support the required activities. Further investigations to evaluate the contribution of group therapy in building specific aspects of resilience and improving mental health should be undertaken. This research could also expand to other regions of Greece with different socio-economic characteristics combined with quantitative research. Finally, it would be interesting to conduct a second study that would include therapists’ perspectives on group therapy experiences.

## Figures and Tables

**Table 1 healthcare-13-01035-t001:** Participants’ socio-demographic characteristics.

Participants’ Code Numbers	Gender	Age	Marital Status	Educational Background	Years of Employment	Health/Social Service
R1	F	45	Married	University	20	Private Sector
R2	F	40	Married	University	15	Private Sector
R3	F	35	Married	University/MA	7	Public Service
R4	F	38	Married	University	8	Private Sector
R5	F	39	Married	High School	6	Public Service
R6	F	44	Married	University	12	Private Sector
R7	F	42	Married	University	15	Private Office
R8	F	43	Married	University	17	Public Service
R9	F	47	Married	University	5	Private Office
R10	F	43	Married	University	11	Public Service
R11	F	45	Married	University/MA	18	Private Office
R12	F	48	Divorced	High School	12	Unemployed

**Table 2 healthcare-13-01035-t002:** The main themes, sub-themes, and key indicators.

Themes	Sub-Themes	Key Indicators
1. Personal development and mental health	1.1. Developing self-confidence	Self-esteem Trust myself Felt secure
	1.2. Handling emotions	Recognizing emotions Keep in touch with feelings Expression of emotions Take care of ourselves
2. Reinforcement of resilience	2.1. Overcome difficulties	Stress management skills Organize things—set priorities
	2.2. Improvement of the relationship with family	New ways of coming in touch with each other in family New mechanism of coping with stress and anxiety in family Realize mistakes with family members
	2.3. New ways of communicating	Respect other people Express myself Cope with quarrels
3. Coping with health crisis	3.1. Self-efficacy	Control fears Mechanisms of managing stress
	3.2. Strategies to overcome daily difficulties	Follow a timetable Stay in touch with my family Communicate via technology
	3.3. COVID-19 as a challenge	COVID-19 as a chance COVID-19 with a positive perspective
	3.4. New ways of behaving	Move forward Collective hope
4. Describing group therapy’s impact through an image	4.1. Images: sense of improvement and continuous development	Difficult road with beautiful roses Opened pomegranate
	4.2. Images: source of power, support, and care giving	Nest

## Data Availability

The data presented in this study are available on request from the corresponding author.

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
