# Peer review of "Promoting Women’s Mental Health and Resilience in Times of Health Crisis and Adversity via Personal Development Groups"

_healthcare, 2025, doi:10.3390/healthcare13091035_

Round 1
Reviewer 1 Report
Comments and Suggestions for Authors
Regarding the manuscript “Promoting women’s mental health and resilience in times of health crisis and adversity via personal development groups”, despite its potential contribute to promoting mental health and resilience, while addressing gender inequalities, as it stands, its publication is not recommended.
The following aspects can contribute to the improvement of the manuscript:
- It is advisable to review the presentation of the aims in the abstract, ensuring consistency throughout the manuscript (e.g., “2.1. Aim”);
- To revise the wording of the manuscript to make it more fluid and coherent. There is also the need for greater theoretical depth in the introduction and discussion of the results. It would also be relevant to reflect on other factors that may have influenced the results;
- It would be important to provide further clarification of the therapy methods (lines 145-146);
- The reference to the small sample size as a study limitation should be revised and removed. However, other limitations of the research should be added and analysed;
- The discussion of the results from this study does not address the future recommendations. This aspect should be integrated and explored in deep in the manuscript;
- As for the final considerations, it is advisable to better explain the significance of the results obtained and their potential contribution to this field of research;
- There are sentences in the manuscript that appear in different font sizes, which should be reviewed.
Author Response
Dear Reviewer 1,
RESPONSE TO THE REVIEWER 1:
We would like to thank the reviewer for the thorough reading of our work and the very constructive comments which contributed significantly to the improvement of the initial manuscript. We tried to respond to all the issues raised.
|
COMMENT |
REPLY |
|
1. It is advisable to review the presentation of the aims in the abstract, ensuring consistency throughout the manuscript (e.g., “2.1. Aim”); |
Please see the changes we made in the abstract and methods.
Abstract (Aim of the study): “This follow-up study examined the viewpoints of women regarding the durability and efficacy of group therapy's positive impact on their resilience and mental health especially through unforeseen circumstances and crises.This group of women had participated in group therapy in order to help them improve their mental health and enhance their resilience” (p. 1, lines: 22-26).
In 2. Methods (2.1. Aim) “The objective of the current follow-up study was to evaluate the perspectives of women who had undergone group therapy from 2012 to 2015 regarding the efficacy of the group in improving their mental health and enhancing their resilience especially during times of unforeseen circumstances and crises.” (p. 3, lines: 124-127) |
|
2. There is also the need for greater theoretical depth in the introduction and discussion of the results. It would also be relevant to reflect on other factors that may have influenced the results; |
Please see the text added to the last paragraph of introduction and to discussion regarding the other factors that are related to resilience.
Introduction: “The ecological approach defines resilience as a dynamic interplay between personal (active coping, reflective functioning, meaningmaking, optimism, flexibility, and hopeful-ness), relational (social support, partner relationships, and family relationships), and contextual (health and social networking, cultural factors, and social policies) factors [23,24,]. The precise role that these factors play in fostering resilience is, in fact, unknown.” (p. 3, lines 116-121)
We have also added in discussion about Discussion: “Certainly, in our study, there is a gap in the way resilience is built, because it is defined as the capacity for managing the availability and utilization of psychological, social and physical resources [23]. According to our findings, group therapy only had an impact on psychological or personal aspects. We had not information about the women’s social and contextual interactions and their role in fostering resilience and promoting general well-being.” ( p. 11, lines: 525-530) |
|
3. The reference to the small sample size as a study limitation should be revised and removed. However, other limitations of the research should be added and analyzed; |
The limitation regarding the small sample size in now removed. We have added other limitations (p.12) |
|
4. The discussion of the results from this study does not address the future recommendations. This aspect should be integrated and explored in deep in the manuscript; |
Future recommendations are now added in Conclusion.
Conclusion: “Group therapy is an effective therapeutic approach that social workers, psychologists, psychotherapists, and other practitioners can utilize combined with individual or family consulting. At both the local and national levels, social policy plays a vital role. It will cover training needs and provide funding to support the required activities. Further investigation to evaluate the contribution of group therapy in building specific aspects of resilience and improving mental health should be undertaken. This research could also expand to other regions of Greece with different socio-economic characteristics combined with quantitative research. Finally, it would be interesting to conduct a second study that would include therapists' perspectives on group therapy experiences.” (p.13, lines:572-580) |
|
5. As for the final considerations, it is advisable to better explain the significance of the results obtained and their potential contribution to this field of research; |
The significance of the results are presented in the first paragraph of the conclusion.
Conclusion: Our results highlight key factors about the role of group therapy in fostering women’s’ mental health and resilience. A possible explanation is that group therapy activates mechanisms and processes that enhance individual’s self-esteem and self-confidence. Furthermore, it contributes to the development of new skills in communicating, handling emotions, solving personal problems and coping with stress. It enhances resilience through active coping, optimism, hopefulness. In addition, our findings add to a growing body of literature related to the contribution of group therapy in coping with the Covid 19 health crisis. (p.12, lines 564-571) |

Reviewer 2 Report
Comments and Suggestions for Authors
Highly Esteemed Authors Maria Moudatsou, Areti Stavropoulou , Michael Rovithis , Dimitrios Mimarakis , Sofia Koukouli
You should receive recognition for their valuable research on women's mental health and resilience through personal development groups during health crises like the Coronavirus pandemic. Page 2 between lines 22-26 contains the detailed explanation of the study's objectives and rationale. However, the following improvements are suggested: The research needs to clearly explain how it addresses current knowledge gaps about the long-term impacts of personal development groups on women's resilience. The linkage presented remains too broad for comprehension and requires further elaboration in the stated section of page 2 (lines 22-26).
The research question needs explicit statement immediately following the rationale to provide clear direction from introduction to methods. The methods section includes sufficient preliminary information as found on page 3 between lines 119 and 124. However, the following improvements are suggested for clarity and reproducibility: The study needs to provide detailed information about the specific selection criteria that identified the 12 participants. Researchers need to clarify how they applied all inclusion or exclusion criteria in their study.
To allow other researchers to replicate the study properly researchers should be given detailed information about the semi-structured interview protocol with included example questions.
The framework analysis procedure needs explicit detailing and researchers must explain their rationale for selecting this approach to support research replicability.
- Statistical analysis and methodology
The study's qualitative approach prevents the use of traditional statistical analysis methods. However, the methodology requires clearer elaboration:
Advisory notes: Researchers must provide a detailed justification for their selection of framework analysis as a qualitative methodology in relation to their research questions (refer to page 3, lines 119-124). The current document lacks tables and figures.
Your document needs to feature an illustrative table that summarizes interview-derived themes while displaying representative quotations to help readers understand and engage with the content.
Visualize connections between identified themes and their role in building resilience through a diagrammatic representation.
- Interpretation and conclusions
Interpretations align with presented data. The conclusions provided on page 2 lines 36-39 successfully emphasize the empowerment aspect of the study. To strengthen your conclusions: 4
Advisory notes:Demonstrate how your research outcomes translate into actionable steps for social policy and healthcare providers as well as community organizations by linking these implications directly to your study conclusions.
Conclusions should explicitly state the exploratory nature of qualitative research to prevent overstating result generalizability.
- Strengths of the manuscript
The manuscript successfully highlights personal development groups as effective tools for building resilience and mental health (page 2, lines 32-37).
Advisory notes: The study's six-year follow-up period represents a key methodological strength that requires clear emphasis as a unique contribution.
- Limitations clearly stated
Limitations are inadequately detailed at this stage.
Advisory notes: The research acknowledges the small sample size of twelve participants as a limitation and discusses its impact on the study's generalizability.
The potential for recall bias needs to be examined given the six-year interval between intervention and follow-up. The manuscript displays generally acceptable structure and flow but still requires enhancements to reach full potential.
To improve readability in the Methods section divide it into distinct subheadings like "Participants," "Procedure," and "Data Analysis." Language EditingThe manuscript would benefit from professional language editing to enhance readability, correct minor grammatical errors, and ensure scholarly style consistency.
Final Recommendation: The manuscript should undergo Minor Revision because it presents original content on an important topic using detailed qualitative analysis. Implementing the proposed enhancements will enhance both the clarity and rigor of your manuscript and amplify its impact on readers.
I truly appreciate the chance to assess your manuscript exploring women's mental health and empowerment which presents vital insights for global health challenges. The study's unique contribution to existing literature is highlighted through the authors' innovative longitudinal methodology which stands out as a testament to their approach.
Yours truly,
Comments on the Quality of English LanguageThe manuscript’s English language remains comprehensible but requires refinement in specific regions to boost clarity and professionalism. Certain sentences require simplification to improve their readability and clarity. The manuscript contains small grammatical mistakes and punctuation errors. I recommend only AUTHOR PROOFREADING - in depth - not professional services - just scan the paper repeatedly and correct minor issues.
Author Response
Dear Reviewer 2,
RESPONSE TO THE REVIEWER 2
We would like to thank the reviewer for the thorough reading of our work and the very constructive comments which contributed significantly to the improvement of the initial manuscript. We tried to respond to all the issues raised.
|
COMMENT |
REPLY |
|
1. Page 2 between lines 22-26 contains the detailed explanation of the study's objectives and rationale. However, the following improvements are suggested: The research needs to clearly explain how it addresses current knowledge gaps about the long-term impacts of personal development groups on women's resilience. The linkage presented remains too broad for comprehension and requires further elaboration in the stated section of page 2 (lines 22-26) |
Please see below the improvements in the methods section according to your suggestions.
2. Methods 2.1. Aim “In our study women’s resilience and mental health was evaluated six years after the completion of group therapy. Therefore, other factors, such as the women’s social and contextual environment might have an influence on these positive outcomes. To address the knowledge gap regarding the long-term impact of personal development groups, the same interview guide was applied as the one used during the initial assessment of the group influence. Additionally, a brief reminder of the thematic units covered in this first evaluation was provided.” (p.3, lines: 127-133) |
|
2. The study needs to provide detailed information about the specific selection criteria that identified the 12 participants. Researchers need to clarify how they applied all inclusion or exclusion criteria in their study. |
Please see below our reply regarding the selection criteria.
2.4. Participants: Women who expressed a desire to enhance their resilience and meet their mental needs attended the personal development group. It was part of a larger project aiming at the evaluation of women’s health needs at the local community level. (p. 5, lines: 192-195) |
|
3. To allow other researchers to replicate the study properly researchers should be givendetailed information about the semi structured interview protocol with included example questions. |
Please see some additional details regarding the interview guide.
2.6. Measures: The interview guide is consisted of the following thematic topics: Personal development group and its impact on mental health, personal development group and resilience, visual metaphor representing the group’s overall contribution to women’s mental health and resilience. Example of research questions are: «What influence does the personal development group have on your self-esteem? Are there skills you have gained from your participation in this personal development group? Would you like to talk about them? How do you think that this personal development group helped you in identifying and controlling your emotions? Would you like to give us some examples? (p.5-6, lines:217-225) |
|
4. The framework analysis procedure needs explicit detailing and researchers must explain their rationale for selecting this approach to support research replicability. |
The framework analysis used is presented below in more details.
2.7. Data analysis: “This is a typical method to manage and analyze vast amounts of data from semi-structured interviews. Multidisciplinary research teams utilize it to create a clear and comprehensive description of research data. It is an appropriate data analysis because it ensures accurate findings by adopting a systematic approach to data management and analysis, creating a clear audit trail throughout the study process. We chose framework analysis for our study due to our diverse team of academics and the vast amount of data deriving from the semistructured interviews.” (p.7, lines: 231-238) |
|
5. Statistical analysis and methodology:· The study's qualitative approach preventsthe use of traditional statistical analysismethods. However, the methodologyrequires clearer elaboration. |
· It is a qualitative research approach. Therefore, we did not use statistical analysis. We have now enriched the methodology section with further details. Please see our replies to previous comments 1-4. |
|
6. Your document needs to feature an illustrative table that summarizes interview-derived themes while displaying representative quotations to help readers understand and engage with the content. |
We have now added a table as per suggestion. |
|
7. Visualize connections between identified themes and their role in building resilience through a diagrammatic representation. |
We have tried to visualize connections between identified themes and their role in building resilience (only regarding our findings related to resilience) through a diagrammatic representation. Please see figure 1. |
|
8. Interpretations align with presented data. The conclusions provided on page 2 lines 36-39 successfully emphasize the empowerment aspect of the study. To strengthen your conclusions: 4 |
Conclusion is now strengthened by adding paragraph two.
“Group therapy is an effective therapeutic approach that social workers, psychologists, psycho-therapists, and other practitioners can utilize combined with individual or family consulting. At both the local and national levels, social policy plays a vital role. It will cover training needs and provide funding to support the required activities. Further investigation to evaluate the contribution of group therapy in building specific aspects of resilience and improving mental health should be undertaken. This research could also expand to other regions of Greece with different socio-economic characteristics combined with quantitative research. Finally, it would be interesting to conduct a second study that would include therapists' perspectives on group therapy experiences.”(p.13, lines: 572-580) |
|
9. Advisory notes: Demonstrate how your research outcomes translate into actionable steps for social policy and healthcare providers as well as community organizations by linking these implications directly to your study conclusions. |
We have changed the entire conclusion please see our response to comment 8. |
|
9. Conclusions should explicitly state the exploratory nature of qualitative research to prevent overstating result generalizability |
We used qualitative research approach in this study. Results cannot be generalized in qualitative research. |
|
10. Strengths of the manuscriptThe manuscript successfully highlights personal development groups as effective tools for building resilience and mental health (page 2, lines 32-37). Advisory notes: The study's six-year follow-up period represents a key methodological strength that requires clear emphasis as a unique contribution. |
We have added this in the strengths of the manuscript (see conclusion). |
|
11. Limitations clearly stated Limitations are inadequately detailed at this stage.Advisory notes: The research acknowledges the small sample size of twelve participants as a limitation and discusses its impact on the study's generalizability. |
samples are the characteristic of this research approach and its findings cannot be generalized. |
|
12. The potential for recall bias needs to be examined given the six-year interval between intervention and follow-up.The manuscript displays generally acceptable structure and flow but still requires enhancements to reach full potential. |
We tried to answer to these issues raised by the reviewer in ‘Limitations of the study’ |
|
13. To improve readability in the Methods section divide it into distinct subheadings like "Participants," "Procedure," and "Data Analysis." Language Editing: The manuscript would benefit from professional language editing to enhance readability, correct minor grammatical errors, and ensure scholarly style consistency. |
Methods are divided in 7 sub-sections.We have checked the whole text very thoroughly to address all these issues raised by the reviewer. We think this last edition is very much improved compared to the initial one. |

Round 2
Reviewer 1 Report
Comments and Suggestions for Authors
Regarding the manuscript “Promoting women’s mental health and resilience in times of health crisis and adversity via personal development groups”, the revisions have enhanced overall quality of the manuscript.
- Abstract: The abstract has been revised, with particular emphasis on the wording of the aims. The abstract provides a summary of the research, including aims, methods, main results and conclusions.
- Introduction: The authors have revised the wording of the introduction. Although, the authors provide an explanation of the concept of resilience, based on the ecological model, and the potential factors that can enhance resilience, these could have been explored in greater depth. Also, the introduction does not include all the relevant references.
- Methods: The methodology has been revised, with the authors adding the inclusion criteria for the study and the data collection procedures. There was also a concern to clarify the thematic topics covered in the interview guide, as well as the method adopted to analyze the data collected.
- Findings: Table 2 has been added to the presentation of the results, in which the authors explain the main themes and sub-themes and exemplar quotes. Since the quotes are already included in the text, it is advisable instead to explain the indicators that guided the definition of the themes/sub-themes. Despite the effort to systematise the factors related to strengthening resilience in figure 1, its potential contribution is unclear. The use of circular shapes is even questionable. Its is recommend that the authors provide a clearer clarification of what motivated the systematisation of information in this particular way. Also, the explanation of the factors in the text does not follow the order in which the circles are organised. This aspect requires further reflection and alignment.
- Discussion/ limitations: The authors engage in a critical discussion of the study’s results and its limitations.
- Conclusions: The authors improved the manuscript through a deeper exploration its conclusions/potential recommendations.
Author Response
REPLY TO REVIEWER 1 – SECOND ROUND
We thank the reviewer for the constructive comments. Please see our replies to the additional points raised in the introduction and findings.
Comment: Regarding the manuscript “Promoting women’s mental health and resilience in times of health crisis and adversity via personal development groups”, the revisions have enhanced overall quality of the manuscript. Introduction: The authors have revised the wording of the introduction. Although, the authors provide an explanation of the concept of resilience, based on the ecological model, and the potential factors that can enhance resilience, these could have been explored in greater depth. Also, the introduction does not include all the relevant references.
Reply: We have now enriched the paragraph referring to resilience (please see lines 118-131 in green color). It is not clear to us what the reviewer means by ‘the introduction does not include all the relevant references’. Reference 23 referring to resilience (last paragraph of introduction) is included in the references list at the end. However, it must be aligned with the rest of the references.
Comment: Findings: Table 2 has been added to the presentation of the results, in which the authors explain the main themes and sub-themes and exemplar quotes. Since the quotes are already included in the text, it is advisable instead to explain the indicators that guided the definition of the themes/sub-themes.
Reply: We removed from the table the interview excerpts and replaced them with the key indicators that guided the definition of themes/sub-themes, as per suggestion.
Comment: Despite the effort to systematise the factors related to strengthening resilience in figure 1, its potential contribution is unclear. The use of circular shapes is even questionable. Its is recommend that the authors provide a clearer clarification of what motivated the systematisation of information in this particular way. Also, the explanation of the factors in the text does not follow the order in which the circles are organised. This aspect requires further reflection and alignment.
Reply: We attempted to visualize the factors related to resilience as they derived from our qualitative research, according to the reviewer’s suggestion in the first round of the review. However, we might have misunderstood and we recognize that circular shapes are not the appropriate way. Therefore, we prefer to keep only the table with the analysis of themes, subthemes and indicators, which, to our view, describes more accurately the results of the present study.